# Differences between 3D isovoxel fat suppression VIBE MRI and CT models of proximal femur osseous anatomy: A preliminary study for bone tumor resection planning

**Choong Guen Chee[1], Hye Won Chung [1◐*], Wanlim Kim[2◐*], Min A. Yoon[1], So Myoung Shin[3], Guk Bae Kim[3]**

**1** Department of Radiology and Research Institute of Radiology, Asan Medical Center, University of Ulsan College of Medicine, Seoul, Republic of Korea, **2** Department of Orthopedic Surgery, Asan Medical Center, University of Ulsan College of Medicine, Seoul, Republic of Korea, **3** Anymedi Inc., Seoul, Korea

◐ These authors contributed equally to this work.
* chung@amc.seoul.kr (HWC); kalhonaaho@naver.com (WK)

**Data Availability Statement:** All relevant data are within the paper and its Supporting Information files.

## Abstract

### Purpose

To evaluate the osseous anatomy of the proximal femur extracted from a 3D-MRI volumetric interpolated breath-hold (VIBE) sequence using either a Dixon or water excitation (WE) fat suppression method, and to measure the overall difference using CT as a reference standard.

### Material and methods

This retrospective study reviewed imaging of adult patients with hip pain who underwent 3D hip MRI and CT. A semi-automatically segmented CT model served as the reference standard, and MRI segmentation was performed manually for each unilateral hip joint. The differences between Dixon-VIBE-3D-MRI vs. CT, and WE-VIBE-3D-MRI vs. CT, were measured. Equivalence tests between Dixon-VIBE and WE-VIBE models were performed with a threshold of 0.1 mm. Bland–Altman plots and Lin's concordance-correlation coefficient were used to analyze the agreement between WE and Dixon sequences. Subgroup analyses were performed for the femoral head/neck, intertrochanteric, and femoral shaft areas.

### Results

The mean and maximum differences between Dixon-VIBE-3D-MRI vs. CT were 0.2917 and 3.4908 mm, respectively, whereas for WE-VIBE-3D-MRI vs. CT they were 0.3162 and 3.1599 mm. The mean differences of the WE and Dixon methods were equivalent ($P$ = 0.0292). However, the maximum difference was not equivalent between the two methods and it was higher in WE method. Lin's concordance-correlation coefficient showed poor

**Funding:** This research was supported by the KIAT (Korea Institute for Advancement of Technology) grant funded by the Korea Government (MOTIE: Ministry of Trade Industry and Energy, specific grant numbers: P0008805). One of our corresponding authors, Wanlim Kim, received the fund. The funders provided support in the form of research grant which was used for the administrative and material support of our study. Funders had no additional role in the study design, data collection and analysis, decision to publish, or preparation of the manuscript. URL to the sponsors' website is "https://www.kiat.or.kr/site/eng/main.jsp". The full name of the commercial company which collaborated with our study is ANYMEDI. Two authors from ANYMEDI, So Myoung Shin and Guk Bae Kim, played their role in study design, data collection and analysis. The specific roles of theses authors are articulated in the 'authors contributions' section. Other than the two authors, So Myoung Shin and Guk Bae Kim, no authors received salary or other funding from ANYMEDI.

**Competing interests:** There are no relative declarations relating to employment, consultancy, patents, products in development, or marketed products. The commercial affiliation, ANYMEDI, does not alter our adherence to PLOS ONE policies on sharing data and materials.

agreement between Dixon and WE methods. The mean differences between the CT and 3D-MRI models were significantly higher in the femoral shaft area ($P$ = 0.0004 for WE and $P$ = 0.0015 for Dixon) than in the other areas. The maximum difference was greatest in the intertrochanteric area for both techniques.

## Conclusion

The difference between 3D-MR and CT models were acceptable with a maximal difference below 3.5mm. WE and Dixon fat suppression methods were equivalent. The mean difference was highest at the femoral shaft area, which was off-center from the magnetization field.

## Introduction

Accurate and precise understanding of the three-dimensional (3D) morphology of osseous anatomy is crucial for surgical planning in orthopedic surgery, and the advent of 3D printing enables the surgeon to physically simulate the surgical procedure with 3D-printed models. Currently, the most accurate method to achieve 3D osseous anatomy for 3D printing is to utilize computed tomography (CT). However, with the advance of magnetic resonance imaging (MRI) technology, there have been numerous efforts to acquire the anatomical detail of osseous anatomy with high resolution 3D-MRI. Black bone MRI sequences [1–5], Dixon 3D Flash MRI sequences [6, 7], and zero-echo-time MRI sequences [8] have all been tested as potential alternatives to 3D-CT, showing promising results. However, previous studies mostly focused on areas of craniofacial bone or intra-articular anatomy, where few tendinous or ligamentous structures insert the bone.

The use of 3D printing in orthopedic surgery is an evolving area; it allows the design of various custom-made prosthesis and patient-specific resection guides for wide resection of bone tumors with a minimal safety margin [9–13]. Park et al. reported a maximal cutting error of 3 mm in a series of 12 patients who underwent orthopedic oncological surgery using a resection guide designed with 3D printing [13]. In their series, the resection guide design was mainly planned using CT imaging, with conventional two-dimensional (2D) MRI being utilized as an aid to evaluate the tumor boundary. However, compared with conventional 2D-MRI, iso-voxel-3D-MRI could provide a higher resolution image without stair-step artifacts in various reformation planes. The more accurate and precise the image, the lower it is possible to make the registration error between the CT and MR images, thereby providing a smaller safety margin. If the osseous anatomy extracted from 3D-MRI is comparable to that of 3D-CT, 3D-MRI may have the potential to substitute 3D-CT when planning the guidance for tumor resection, thereby avoiding the radiation hazard issues of CT. However, there is no previous literature providing the error margin of 3D-MRI in comparison with CT.

The purpose of our paper is to evaluate the 3D osseous anatomy of the proximal femur extracted from 3D-MRI volumetric interpolated breath-hold examination (VIBE) sequences, and to measure the overall differences using CT as the reference standard. The secondary endpoint is to optimize the VIBE sequence by evaluating any differences between Dixon and water excitation (WE) fat suppression techniques.

## Material and methods

Institutional review board of Asan Medical Center approved this single-center retrospective study, and the requirement for informed patient consent was waived because of its retrospective nature.

### Patient selection

A retrospective review of patients who underwent pelvis MRI between May 15th and July 1st, 2019 because of hip or inguinal pain was performed. In addition to the routine sequence, the patients also underwent a 3D-MRI protocol, with the field of view (FOV) designed to cover a single hip joint. Adult patients who underwent 3D pelvis MRI and pelvis CT within a 1-month interval, and who had not undergone prior hip surgery, were included in this study. The side of the hip joint with normal anatomy, which did not have femoral head collapse, deformity, or advanced osteoarthritis (Tönnis grade 2 or 3), was selected according to the hip anteroposterior radiography, which was obtained before the pelvis MRI examination. The exclusion criteria were i) patients whose 3D-MRI was not acceptable because of motion artifact, ii) patients who had femoral collapse, deformity, or advanced osteoarthritis of both hip joints, and iii) patients who needed sedation.

### MRI and CT protocols

All MRI studies were performed on a 3-T MRI scanner (Skyra; Siemens Healthineers, Erlangen, Germany). In addition to the routine sequences, volumetric imaging using a T1-weighted fast 3D gradient-echo VIBE sequence, was obtained. Imaging of a unilateral hip was obtained in the coronal plane with the femoral head centered in a 230 mm FOV (Fig 1). The images were obtained with the combined use of an 18-channel anterior body coil and 32-channel spine coil. To maintain the image quality for axial and sagittal multiplanar reformations, images were obtained with a 0.9 mm isovoxel resolution (FOV: 230 mm, Matrix size: $256 \times 256$, slice thickness: 0.9 mm; right to left phase encoding direction). Two different fat suppression techniques (Dixon and WE methods) were used. The parameters for the Dixon method included TR: 5–6 ms, $TE_2$: 3–4 ms, $TE_1$: 2–3 ms, flip angle: 10.0˚, and lines per shot: 240; those for the WE included TR: 7–8 ms, TE: 4–5 ms, flip angle: 10.0˚, and lines per shot: 20. No contrast agent was administered for the examination. The imaging acquisition time of the Dixon-VIBE sequence was 5 min 50 s, whereas that of the WE-VIBE sequence was 6 min 40 s.

CT studies were performed on various 64-slice CT scanners (Siemens Healthineers, Erlangen, Germany; GE Healthcare, Chicago, IL, USA), with coverage from the upper-most iliac crest to the proximal diaphysis of the femur. Pelvis CT was acquired with a tube voltage of 120 kV, X-ray tube current-exposure time product within a range of 100–250 mAs, collimation of either 0.6 or 0.625 mm, beam pitch within a range of 0.8–1, and a smooth reconstruction algorithm.

### Segmentation and 3D-MRI-CT registration

Segmentation of the osseous anatomy of the proximal femur on 3D-CT was performed using a semi-automated method with a high-pass filter and a threshold value of 190 Hounsfield units. MRI segmentation was performed manually by a radiologist cooperating with a segmentation specialist. A low pass filter was applied to the fat suppressed image to extract the low signal areas that encompassed the bone, fatty tissues, tendons, and ligaments (Fig 2A). For the Dixon-VIBE images, the outlining boundary of the cortex or subchondral bone plate was

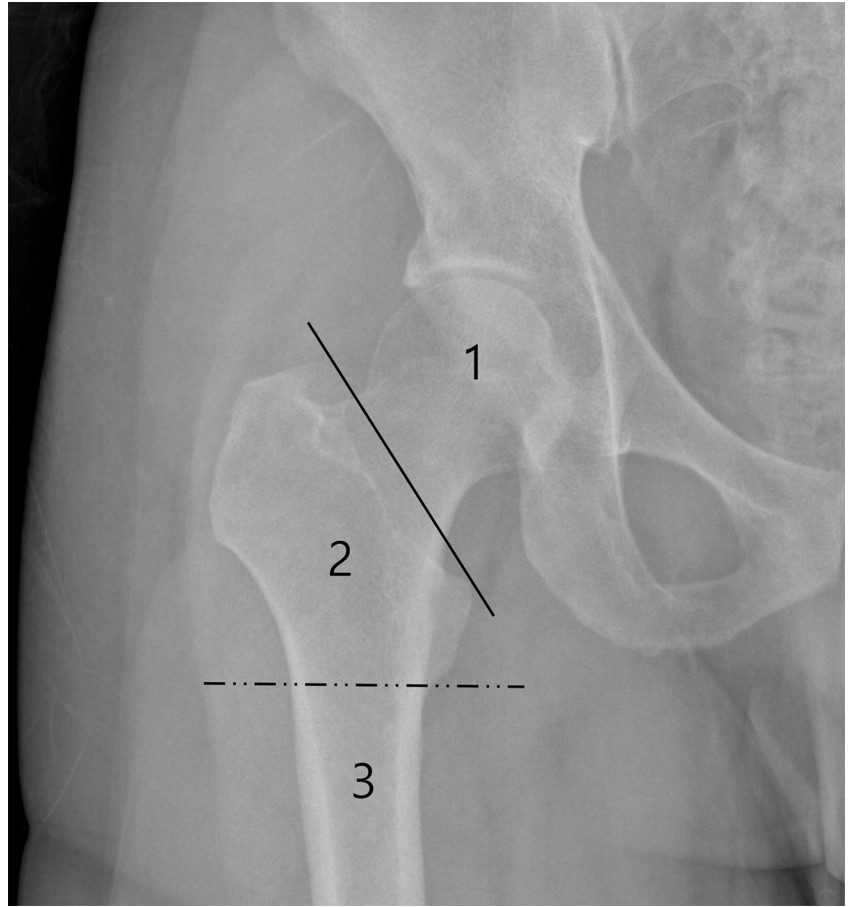

**Fig 1. The field of view (FOV) of the 3D-MRI.** The unilateral hip joint was covered by a 230 × 230 mm FOV in the coronal plane. The proximal femur was divided into epi-metaphyseal (area 1: femoral head and neck area), metaphysis (area 2: intertrochanteric area), and diaphysis areas (area 3: femoral shaft area).

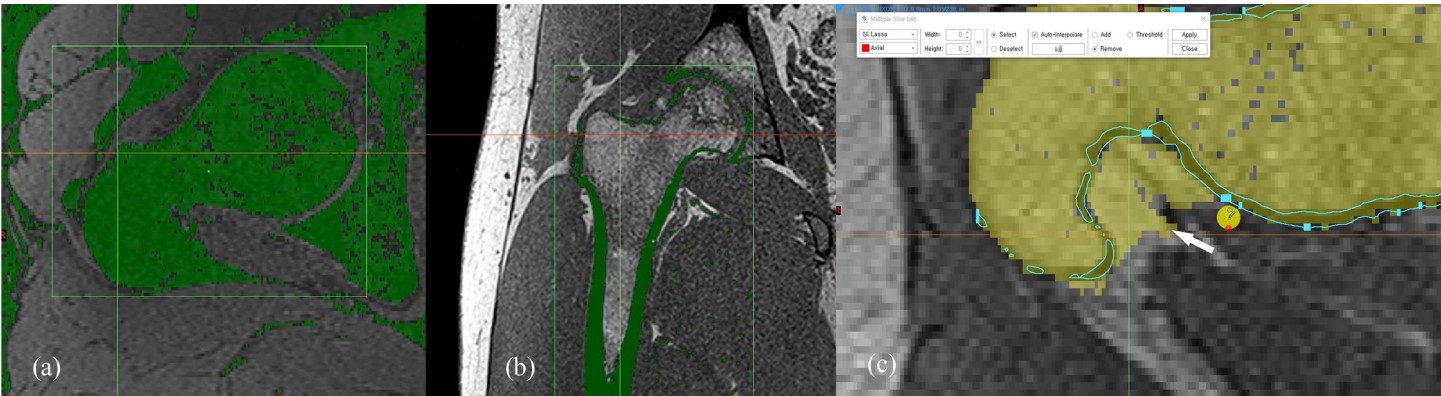

**Fig 2. Manual segmentation process of 3D-MRI model.** Using a low pass filter, bone structure can be extracted from the fat suppressed MR image, along with fatty tissues, tendons, and ligaments (a). Using the in-phase image obtained in the Dixon method, the bony cortex or subchondral bone plate can be extracted by using low pass filter (b), unlike with the water excitation method. The extracted boundary of the cortex or subchondral bone plate served as a guideline (blue line) between the osseous anatomy and overlying subcutaneous fat (white arrow) during the bone segmentation (c).

extracted by applying a low pass filter to the fat-only or in-phase images (Fig 2B). The extracted boundary of the cortex or subchondral bone plate served as a guideline between the osseous anatomy and overlying subcutaneous fat (Fig 2C). In certain areas where tendon attached, such as the greater trochanter, the boundary between the tendon and bone was determined by visual assessment by the radiologist. In the WE fat suppression method, the boundary between the osseous structures and overlying subcutaneous fat or tendon was judged by the radiologist without a guideline, unlike in the Dixon technique. A 2-week interval was maintained between the Dixon-based 3D-MRI and WE-based 3D-MRI bone segmentations, to avoid recall bias. The Materialise Mimics program (Materialise, Leuven, Belgium) was used for the segmentation process.

The stereolithography (STL) file of the 3D-CT semi-automatically segmented proximal femur model served as a reference for the 3D-MRI segmentation models of the proximal femur. A global registration method was used to register the Dixon-based 3D-MRI segmented bone model and the WE-based 3D-MRI segmented proximal femur model to the 3D-CT bone model. Global registration was applied over multiple iterations until the average distance error of the registrations between the CT and MRI-based 3D femur models reached a nadir. After reaching the optimal global registration between the CT and MRI-based femur models, the mean, standard deviation, and maximum error of the differences between the two models were measured for each patient. Difference maps were calculated to portray the differences between the 3D-MRI segmented femur models and the reference 3D-CT segmented bone model using color mapping (Fig 3). To simulate the segmentation of bone tumors involving epiphyseal, metaphyseal, and diaphyseal areas, the proximal femur was subdivided into femoral head, intertrochanteric, and femoral shaft areas. For the subgroup analysis, the global registration processes were performed separately for the three subdivided areas. The total time needed for the segmentation process was recorded. The registrations and difference measurements between the CT and MRI models were analyzed using the Materialise 3-matic program (Materialise, Leuven, Belgium).

## Statistical analysis

The mean, maximum, and standard deviation of the differences between the CT and Dixon-VIBE 3D-MRI and WE-VIBE 3D-MRI models were measured. The measurements were repeated for the three subregions of the proximal femur. Equivalence testing was performed with a threshold of 0.1 mm to evaluate whether the differences between Dixon-VIBE 3D-MRI and CT, and between WE-VIBE 3D-MRI and CT, were equivalent, with the CT model regarded as the reference standard. Additionally, Bland–Altman plots were used to assess the extent of measured differences between the CT and Dixon-VIBE 3D-MRI and WE-VIBE 3D-MRI. Linear mixed model analysis was performed to investigate whether there was a regional difference between the 3D MRI and CT models. Student's *t*-test was performed to evaluate differences in measurements according to the osteonecrosis disease status of the femoral head. Pearson correlation analysis was used to evaluate correlations between bone mineral density and the measurement difference between the 3D MRI and CT models. P-values of <0.05 were interpreted as statistically significant. All statistical analyses were performed using SAS version 9.4 (SAS Institute, Cary, NC, USA).

## Results

Of the 20 patients initially identified as being possibly eligible, three were excluded because of motion artifact, one because of bilateral post-collapse osteonecrosis of the femoral head, and one because of a requirement for sedation. Therefore, 15 patients were analyzed in this study (mean age, 50 years ± 18 [standard deviation]; 10 men), and their characteristics are summarized in Table 1. The mean, maximum, and standard deviation of the differences between the

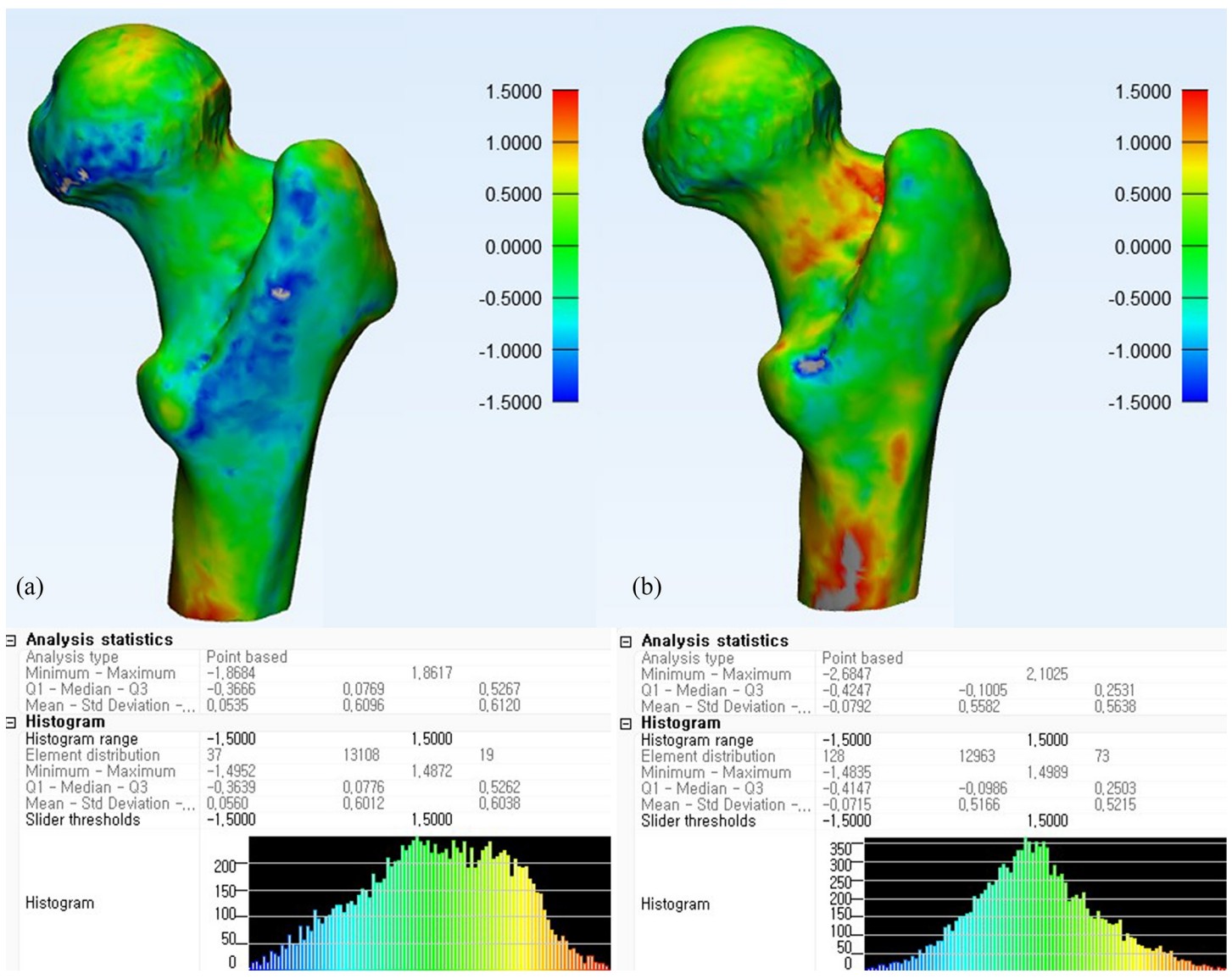

**Fig 3. Difference map between 3D-MRI models and CT of a proximal femur.** Difference maps of a 20-year-old female proximal femur with normal osseous anatomy using Dixon (a) and water excitation methods (b).

**Table 1. Patient characteristics.**

| Characteristics | |
|---|---|
| Total number of included patients | 15 |
| Number of normal hips | 9 |
| Number of hips showing osteonecrosis of femoral head (Ficat stage IIA) | 6 |
| Age (years), mean ± standard deviation | 50 ± 18 |
| Sex (M:F) | 10:5 |
| Bone mineral density, spine, mean ± standard deviation | 1.08 ± 0.14 |
| Bone mineral density, femur, mean ± standard deviation | 0.92 ± 0.13 |

**Table 2. Image registration error for the total hip between 3D-CT and 3D-MRI using the water excitation method.**

| Case | Hip status | BMD (spine) | BMD (femur) | Mean (mm) | SD (mm) | Q1 (mm) | Median (mm) | Q3 (mm) | Max value (mm) | Segmentation time (min) |
|---|---|---|---|---|---|---|---|---|---|---|
| case 1 | normal | 0.969 | 0.833 | 0.197 | 0.543 | −0.147 | 0.264 | 0.577 | 1.942 | 23 |
| case 2 | ONFH | 0.99 | 0.905 | 0.413 | 0.830 | −0.148 | 0.293 | 0.971 | 3.446 | 22 |
| case 3 | ONFH | 0.843 | 0.988 | 0.034 | 0.660 | −0.267 | 0.076 | 0.415 | 4.167 | 38 |
| case 4 | ONFH | 1.177 | 1.101 | 0.330 | 0.664 | −0.118 | 0.287 | 0.796 | 2.812 | 50 |
| case 5 | normal | 1.21 | 1.171 | 0.346 | 0.865 | −0.273 | 0.224 | 0.925 | 3.568 | 48 |
| case 6 | ONFH | 1.027 | 0.714 | 0.279 | 0.830 | −0.303 | 0.136 | 0.870 | 2.755 | 42 |
| case 7 | normal | 1.123 | 0.852 | 0.565 | 0.949 | −0.075 | 0.504 | 1.121 | 4.304 | 56 |
| case 8 | normal | 1.279 | 1.032 | 0.292 | 0.844 | −0.275 | 0.211 | 0.843 | 2.887 | 47 |
| case 9 | normal | 0.889 | 0.704 | 0.268 | 0.929 | −0.362 | 0.202 | 0.890 | 3.202 | 44 |
| case 10 | normal | 1.086 | 0.925 | 0.215 | 0.598 | −0.151 | 0.258 | 0.631 | 2.815 | 47 |
| case 11 | ONFH | 1.219 | 0.963 | 0.186 | 0.681 | −0.273 | 0.171 | 0.666 | 2.716 | 69 |
| case 12 | normal | 1.138 | 0.946 | 0.114 | 0.453 | −0.164 | 0.114 | 0.388 | 2.403 | 33 |
| case 13 | normal | 1.287 | 1.061 | 0.318 | 0.892 | −0.081 | 0.215 | 0.592 | 5.529 | 57 |
| case 14 | ONFH | 1.039 | 0.815 | 0.396 | 1.158 | −0.209 | 0.254 | 0.851 | 6.802 | 51 |
| case 15 | normal | 0.957 | 0.842 | 0.423 | 0.933 | −0.223 | 0.254 | 1.153 | 3.015 | 56 |
| Average | | 1.08 | 0.92 | 0.292 | 0.789 | −0.205 | 0.231 | 0.779 | 3.491 | 46 |

BMD: bone mineral density; ONFH: osteonecrosis of femoral head; SD: standard deviation; Q1: lower quartile; Q3: upper quartile.

3D-MRI and 3D-CT models were 0.292, 3.491, and 0.789 mm, respectively, for Dixon-VIBE (Table 2), and 0.316, 3.160, and 0.680 mm for WE-VIBE (Table 3). The mean differences between the 3D-CT and 3D-MRI models were considered equivalent in regard to the two fat suppression techniques; however, the maximum value of the differences was higher in the WE method than in the Dixon-VIBE method (3.491 vs. 3.160 mm; Table 4). The Bland–Altman plots showed points scattered above and below zero, suggesting there was no consistent bias

**Table 3. Image registration error for the total hip between 3D-CT and 3D-MRI using the Dixon method.**

| Case | Hip status | BMD (spine) | BMD (femur) | Mean (mm) | SD (mm) | Q1 (mm) | Median (mm) | Q3 (mm) | Max value (mm) | Segmentation time (min) |
|---|---|---|---|---|---|---|---|---|---|---|
| case 1 | normal | 0.969 | 0.833 | 0.105 | 0.559 | −0.233 | 0.122 | 0.445 | 2.859 | 26 |
| case 2 | ONFH | 0.99 | 0.905 | 0.435 | 1.007 | −0.211 | 0.303 | 1.039 | 3.880 | 38 |
| case 3 | ONFH | 0.843 | 0.988 | 0.407 | 0.553 | 0.054 | 0.385 | 0.759 | 2.877 | 50 |
| case 4 | ONFH | 1.177 | 1.101 | 0.330 | 0.729 | −0.219 | 0.271 | 0.845 | 2.829 | 50 |
| case 5 | normal | 1.21 | 1.171 | 0.218 | 0.993 | −0.415 | 0.076 | 0.670 | 4.138 | 31 |
| case 6 | ONFH | 1.027 | 0.714 | 0.224 | 0.488 | −0.124 | 0.200 | 0.564 | 2.107 | 43 |
| case 7 | normal | 1.123 | 0.852 | 0.440 | 0.670 | 0.060 | 0.394 | 0.781 | 3.575 | 34 |
| case 8 | normal | 1.279 | 1.032 | 0.432 | 0.913 | −0.200 | 0.348 | 0.997 | 3.871 | 54 |
| case 9 | normal | 0.889 | 0.704 | 0.178 | 0.622 | −0.184 | 0.164 | 0.494 | 4.725 | 47 |
| case 10 | normal | 1.086 | 0.925 | 0.220 | 0.585 | −0.138 | 0.221 | 0.576 | 2.449 | 30 |
| case 11 | ONFH | 1.219 | 0.963 | 0.397 | 0.621 | −0.023 | 0.384 | 0.821 | 2.506 | 62 |
| case 12 | normal | 1.138 | 0.946 | 0.285 | 0.468 | −0.042 | 0.265 | 0.596 | 2.252 | 33 |
| case 13 | normal | 1.287 | 1.061 | 0.320 | 0.608 | −0.052 | 0.256 | 0.644 | 2.879 | 48 |
| case 14 | ONFH | 1.039 | 0.815 | 0.409 | 0.838 | −0.147 | 0.303 | 0.919 | 4.152 | 50 |
| case 15 | normal | 0.957 | 0.842 | 0.344 | 0.549 | −0.004 | 0.301 | 0.658 | 2.300 | 52 |
| Average | | 1.08 | 0.92 | 0.316 | 0.680 | −0.125 | 0.266 | 0.721 | 3.160 | 43 |

BMD: bone mineral density; ONFH: osteonecrosis of femoral head; SD: standard deviation; Q1: lower quartile; Q3: upper quartile.

**Table 4. Differences between the 3D-CT and 3D-MRI models with two different fat suppression methods (Dixon and water excitation).**

| | Region | Water excitation | Dixon | p-value |
|---|---|---|---|---|
| | | Mean ± Standard deviation | Mean ± Standard deviation | |
| Mean difference between 3D-MRI and 3D-CT model | Total area | 0.292 ± 0.113 | 0.316 ± 0.107 | *0.029 |
| | Femoral head area | 0.307 ± 0.178 | 0.284 ± 0.167 | *0.027 |
| | Intertrochanteric area | 0.232 ± 0.127 | 0.292 ± 0.102 | 0.0995 |
| | Femur shaft area | 0.441 ± 0.192 | 0.499 ± 0.166 | 0.2093 |
| Maximal difference between 3D-MRI and 3D-CT model | Total area | 3.491 ± 1.270 | 3.160 ± 0.826 | 0.7647 |
| | Femoral head area | 2.292 ± 0.997 | 1.989 ± 0.440 | 0.8271 |
| | Intertrochanteric area | 2.650 ± 0.768 | 2.517 ± 0.705 | 0.5635 |
| | Femur shaft area | 2.276 ± 0.682 | 2.315 ± 0.933 | 0.4072 |
| Standard deviation of difference between 3D-MRI and 3D-CT models | Total area | 0.789 ± 0.185 | 0.680 ± 0.177 | 0.5680 |
| | Femoral head area | 0.543 ± 0.264 | 0.484 ± 0.084 | 0.2647 |
| | Intertrochanteric area | 0.596 ± 0.099 | 0.5550 ± 0.107 | 0.0506 |
| | Femur shaft area | 0.679 ± 0.176 | 0.630 ± 0.206 | 0.1901 |

* $p < 0.05$, taken to indicate equivalence between the water excitation and Dixon methods.

between the Dixon and WE methods (Fig 4). Except for one outlier, the points were located within 1.96 standard deviations, which suggests that the agreement between the two fat suppression methods were acceptable.

Linear mixed model analysis showed that the mean difference was highest at the femoral shaft area, with it being significantly higher than at the femoral head and neck and intertrochanteric areas for both WE ($p < 0.001$) and Dixon ($p < 0.0015$) methods. Geographical distortion of the MRI models was observed at the femoral shaft, which was located considerably off-center from the magnetization field (Fig 5). The maximum difference was highest in the intertrochanteric area for both techniques; however, intertrochanteric area was significantly higher than other areas only with the Dixon technique ($p < 0.0201$).

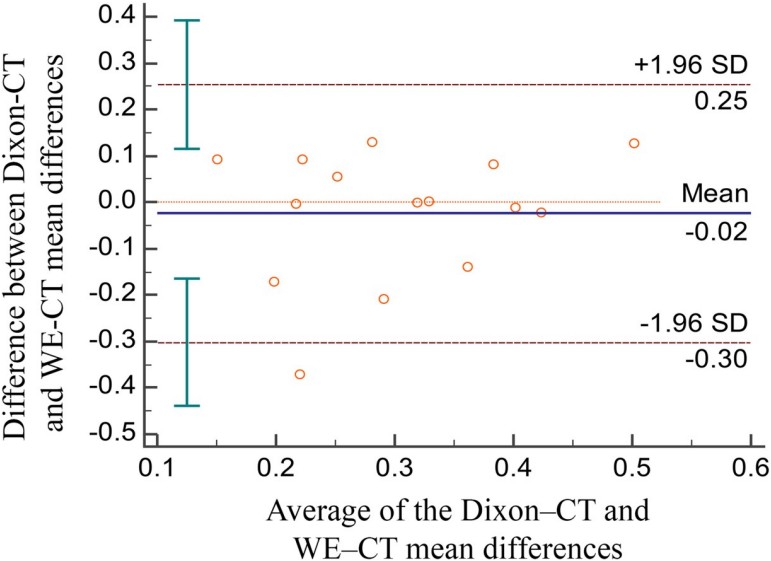

**Fig 4. Bland–Altman plots.** Bland–Altman plots comparing the mean difference of WE-VIBE 3D-MRI vs. CT with the mean difference of Dixon-VIBE 3D-MRI vs. CT.

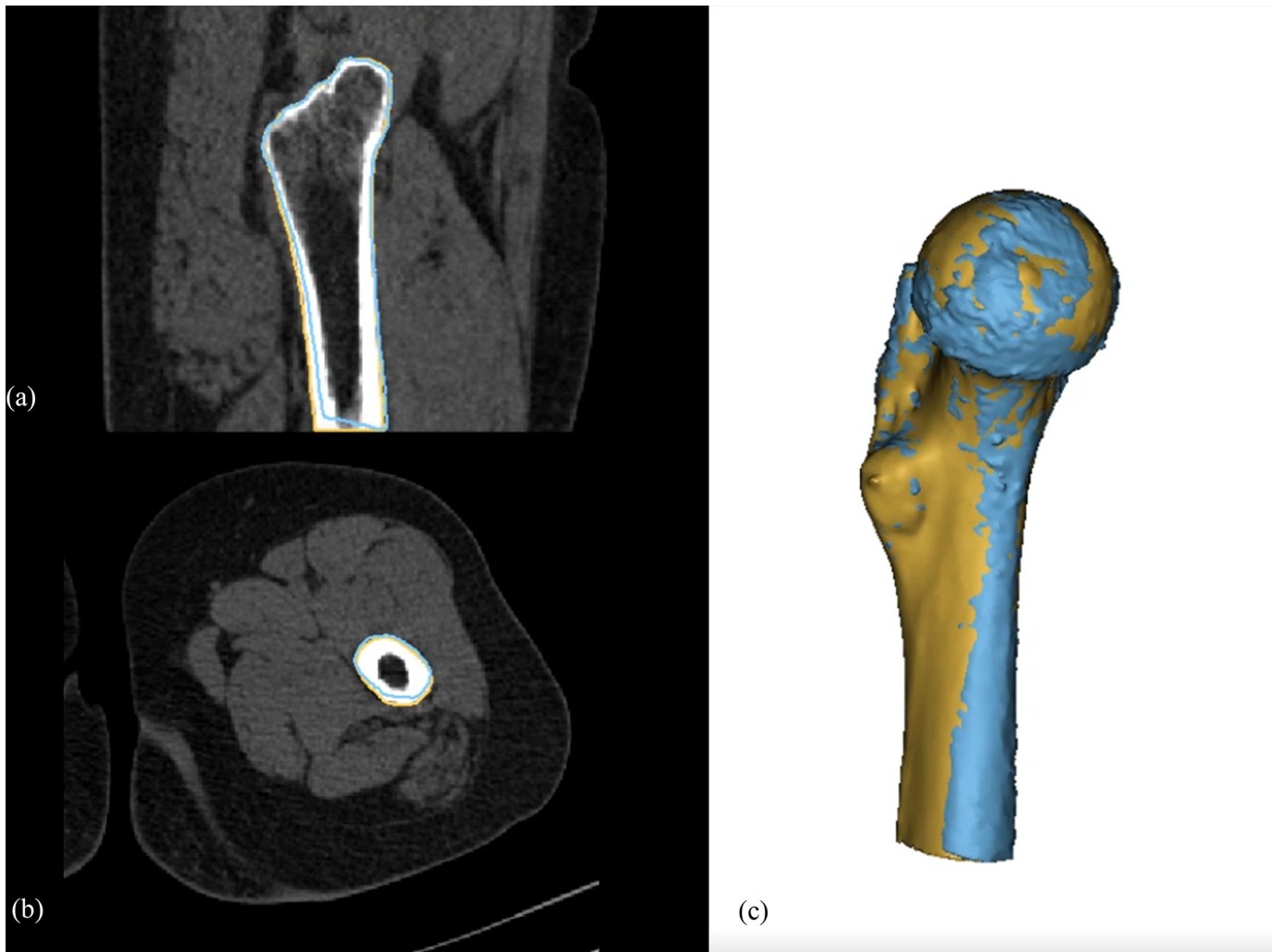

**Fig 5. Distortion of MR model at the periphery of the magnetic field.** Sagittal (a), axial (b), and 3D-modelling images (c) of a proximal femur in a 32-year-old male patient. The blue line indicates the bone contour achieved from the Dixon-VIBE-3D-MRI model, and the brown line indicates that of the CT model. The anterior distortion of the MRI model in respect to the CT model is aggravated at the periphery of the magnetic field (femur shaft area).

Bone mineral density of the femur showed poor correlation with the differences between the 3D-MRI and 3D-CT models (Pearson correlation coefficient = 0.22219; p = 0.4261), as did the bone mineral density of the spine (Pearson correlation coefficient = 0.18634; p = 0.5061). The measurement differences did not significantly differ according to the presence of osteonecrosis, with either WE (p = 0.6729) or Dixon (p = 0.1369) methods. The average times needed for segmentation of a proximal 3D-MRI femur model were 46 min (range: 22–69 min) and 43 min (range: 26–62 min) for WE and Dixon methods, respectively.

## Discussion

Isovoxel 3D-MRI VIBE sequences with WE and Dixon fat suppression methods showed mean differences from the 3D-CT model of 0.292 and 0.3162 mm, respectively. The two different fat suppression methods were equivalent with a threshold of 0.1 mm. The mean differences between the 3D MRI and CT models were most notable at the femoral shaft area, which was placed at the most peripheral area of the magnetic field. Distortion of MR images at the edges

of the magnetic field was previously reported by Jafar et al. using a 3D-printed grid phantom [14], and such distortion was also observed in our study (**Fig 5**), causing systematic errors by distorting MRI models anteriorly at the edge of the magnetic field. The effect of distortion on the MRI model was also observed in the subgroup analysis. The maximal difference was greater when the registration process was performed on the entire proximal femur model rather than on each individual subgroup area. As the purpose of the registration algorithm is to minimize the average errors between the 3D CT and MRI models, this disproportional distortion effect, which increases as the object is placed further away from the center of the magnetic field, is most apparent when the model length is long and located in an area toward the edge of the magnetic field. These effects are likely to be more exaggerated if the FOV is >230 mm.

Suppression of the bone marrow signal was necessary to avoid chemical artifacts, which may be problematic especially in the femoral head area. The thickness of subchondral bone plate of femoral head measures below 0.9mm [15], which would be smaller than the size of a voxel in our study protocol. Therefore, unlike femoral shaft area, the chemical shift artifact occurring between the interface of cartilage and fatty marrow would influence the geometry of the femoral head. Referring to the previous report that Dixon method was more effective in fat suppression and yielded higher signal to noise ratio compared to chemical shift-selective technique for 3D T1-weighted MR imaging [16], the authors decided WE method as a comparative arm for Dixon fat suppression method.

The maximal difference between the 3D MRI and CT models were 3.491 and 3.160 mm for WE and Dixon methods, respectively, which are within the range of those in the study by Jafar et al. [14]. The maximal difference was greatest in the intertrochanteric area. The segmentation of this area took the longest time because there are various tendons attaching to the greater and lesser trochanter. The boundary between the bony cortex and tendon was not easily differentiable by signal intensity, and the border between the two structures was determined by the radiologist's visual interpretation based on their understanding of the proximal femur osseous anatomy. The Dixon method has an advantage over the WE method in that it can visualize the boundary by utilizing the in-phase image (**Fig 1B**), and this may be responsible for the difference between the two methods, especially in the intertrochanteric area. The use of zero TE and ultrashort TE imaging may enable differentiation of the cortex from tendon, and these techniques would be promising if they become available with high signal-to-noise ratio and resolution in the future [17].

The overall measurement error in the manufacture of a resection guide would be the sum of the image segmentation error, image registration error, 3D printing error of the resection guide, and error occurring during the surgical process. The determination of an adequate surgical margin considering technical errors, as well as the biological aggressiveness of the tumor, is important. If we could reduce the technical errors, more normal bone could be saved during surgery. To address this issue, our study focused on errors occurring in the first two steps. The global mean difference between the 3D MRI and CT models approximated to 0.3 mm, which seems like a promising result. However, the systematic error due to the distortion of the 3D-MRI bone model at the periphery of the magnetization field, and the average maximal difference of around 3.5 mm, especially in the area of bony prominences where tendons insert, may result in an imperfect fit of the resection guide in the real surgical field. Therefore, our study does not support the replacement of CT with 3D-MRI when designing a resection guide. Rather, we suggest that the resection guide model is designed with CT, which is free from the aforementioned issues, and that the tumor boundary can then be decided on by registration of a 3D-MRI model to the CT model, with an average maximal error of 3.5 mm being taken into

consideration. More importantly, when performing MRI to design a tumor resection guide, the region of interest should be located in the center of the magnetization field.

There are several limitations to our study. First, our study was performed on a small number of models by a single radiologist and single segmentation specialist. The overall segmentation process greatly depends on an understanding of the osseous anatomy and the experience of a segmentation specialist. Therefore, our study results may not be generalized to other osseous anatomy and other institutions. Second, we only performed our study on a normal proximal femur model. Although the incidence of malignant bone tumor is highest around the knee joint, we performed our preliminary study in the hip joint, where we could pair the MRI and CT data. Additionally, more anatomically complex areas such as scapula or pelvic bone would be more technically difficult to segment, and the difference between the 3D MRI and CT models may be higher. Third, we used the CT model as a gold standard. Although CT is generally regarded as the gold standard for 3D printing, the CT segmentation process also needs manual corrections that involve subjective interpretation, and it is still a proxy to a "real" proximal femur. Furthermore, error may occur in the thresholding of Hounsfield units used in the semi-automatic segmentation.

## Conclusion

The difference between 3D-MR and CT models were acceptable (mean difference [Dixon] 0.2917 mm, mean difference [WE] 0.316mm) with a maximal difference below 3.5mm. The WE and Dixon fat suppression methods were equivalent. The mean difference was highest at the femoral shaft area, which was off-center from the magnetization field. Although these small mean errors may be acceptable when designing a patient-specific resection guide, surgeons should obtain images with the required anatomy being centered in the magnetization field because of higher image distortion at the periphery of the field.

## Supporting information

**S1 Table.**
(XLSX)

**S2 Table.**
(XLSX)

## Acknowledgments

The authors appreciate Division of Biostatistics of Asan Medical Center for assistance in statistical analysis.

## Author Contributions

**Conceptualization:** Choong Guen Chee, Hye Won Chung, Wanlim Kim, Min A. Yoon, So Myoung Shin, Guk Bae Kim.

**Data curation:** Choong Guen Chee, Wanlim Kim, Min A. Yoon, So Myoung Shin, Guk Bae Kim.

**Formal analysis:** Choong Guen Chee, Min A. Yoon, So Myoung Shin.

**Funding acquisition:** Wanlim Kim.

**Investigation:** Choong Guen Chee, Hye Won Chung, Min A. Yoon, So Myoung Shin, Guk Bae Kim.

**Methodology:** Choong Guen Chee, Hye Won Chung, Wanlim Kim, Min A. Yoon, Guk Bae Kim.

**Project administration:** Hye Won Chung, Wanlim Kim, Guk Bae Kim.

**Resources:** Guk Bae Kim.

**Software:** So Myoung Shin, Guk Bae Kim.

**Supervision:** Hye Won Chung, Wanlim Kim, Guk Bae Kim.

**Validation:** Choong Guen Chee, Hye Won Chung, Wanlim Kim, Min A. Yoon, So Myoung Shin.

**Visualization:** Choong Guen Chee, Hye Won Chung, Min A. Yoon, So Myoung Shin.

**Writing – original draft:** Choong Guen Chee, Hye Won Chung, Wanlim Kim, Min A. Yoon, So Myoung Shin, Guk Bae Kim.

**Writing – review & editing:** Hye Won Chung, Wanlim Kim.

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
