## [Decision Letter · Decision Letter 0]

20 Jan 2021

PONE-D-20-21489

Differences between 3D isovoxel fat suppression VIBE MRI and CT models of proximal femur osseous anatomy: A preliminary study for bone tumor resection planning

PLOS ONE

Dear Dr. Chung,

Thank you for submitting your manuscript to PLOS ONE. After careful consideration, we feel that it has merit but does not fully meet PLOS ONE’s publication criteria as it currently stands. Therefore, we invite you to submit a revised version of the manuscript that addresses the points raised during the review process.

We are requesting a revision focused on a better tie in between the data presented and tghe conclusions drawn in the manuscript.

We look forward to receiving your revised manuscript.

Kind regards,

JJ Cray Jr., Ph.D.

Academic Editor

PLOS ONE

Journal Requirements:

3. Please amend your Data availability statement to state where the data can be found, and how these can be obtained by other researchers. We note for instance that no Supplemental information has been provided.

4. Thank you for stating the following in the Financial Disclosure section:

'Wanlim Kim

P0008805

Korea Institute for Advancement of Technology

https://kiat.or.kr/site/engnew/index.jsp

The funders had no role in study design, data collection and analysis, decision to publish, or preparation of the manuscript.'

We note that one or more of the authors are employed by a commercial company: Anymedi Inc

Reviewers' comments:

Reviewer's Responses to Questions

**Comments to the Author**

1. Is the manuscript technically sound, and do the data support the conclusions?

Reviewer #1: Yes

2. Has the statistical analysis been performed appropriately and rigorously? 

Reviewer #1: Yes

3. Have the authors made all data underlying the findings in their manuscript fully available?

Reviewer #1: No

4. Is the manuscript presented in an intelligible fashion and written in standard English?

Reviewer #1: Yes

5. Review Comments to the Author

Reviewer #1: The use of 3D printing in orthopedic surgery is an evolving but exciting area. Compared with conventional 2D MRI, isovoxel-3D-MRI could provide a higher resolution image without stair-step artifacts. If the osseous anatomy extracted from 3D-MRI is comparable to that of 3D-CT, 3D-MRI would have the potential to substitute 3D-CT when planning the guidance for tumor resection, thereby avoiding the radiation hazard issues of CT. This study evaluated the 3D osseous anatomy of the proximal femur extracted from 3D-MRI VIBE sequences, and measured the overall differences using CT as the reference standard. The results demonstrated that the difference between 3D-MR and CT models were acceptable with a maximal difference below 3.5mm. The WE and Dixon fat suppression methods were equivalent according to Bland–Altman plots.

This study is quite interesting to read and the conclusion is quite strong due to its experiments. Here, I have several suggestions for the authors.

1. Abbreviations, such as VIBE, should be explained at the first time it appears in your article. You can put it in your title, but for both abstract and you manuscript, you should give the full text.

2. But Dixon and WE based fat suppression techniques are not based on the saturation of the fat. It is not proper to call them fat saturation techniques.

3. Figures are quite low quality for the authors to see clearly. For example, the figure 2, the text in sub-figures can not be seen clearly at all.

4. The authors should give explanation for why fat suppression is necessary for the recon of the bone. Also the 3D osseous anatomy of the proximal femur extracted without fat suppression should be given as a baseline for comparison.

5. Dixon based fat suppression are totally different from the WE in principle. So, it is quite interesting to figure out why they are equal in the discussion, from the medical physics view. It is expected that the authors would give more evidences, such as other reports.

6. References 14 and 15, seems some typos of the authors' name.

6. PLOS authors have the option to publish the peer review history of their article (what does this mean?). If published, this will include your full peer review and any attached files.

Reviewer #1: No

---

## [Author Response · Author response to Decision Letter 0]

14 Mar 2021

1. The PLOS ONE style is applied on the mansucript.

2. Our original mansucript was edited by "Bioedit", which provides a professional scientific editing service. This resubmission was a minor revision with few word changes, and the authors thought that additional editing might not be required. If this matters, please let us know. 

3. Supplemental information is now provided and attached to the resubmission online system. 

4. Financial Disclosure is now stated at the cover letter as follows

* Funding Statement

This research was supported by the KIAT (Korea Institute for Advancement of Technology) grant funded by the Korea Government (MOTIE: Ministry of Trade Industry and Energy, specific grant numbers: P0008805). One of our corresponding authors, Wanlim Kim, received the fund. The funders provided support in the form of research grant which was used for the administrative and material support of this study. Funders had no additional role in the study design, data collection and analysis, decision to publish, or preparation of the manuscript. URL to the sponsors’ website is “https://www.kiat.or.kr/site/eng/main.jsp”. 

The full name of the commercial company which collaborated with our study is ANYMEDI. Two authors from ANYMEDI, So Myoung Shin and Guk Bae Kim, played role in study design, data collection and analysis. The specific roles of theses authors are articulated in the ‘authors contributions’ section. Other than the two authors, So Myoung Shin and Guk Bae Kim, no authors received salary or other funding from ANYMEDI.

* Competing Interests Statement

There are no relative declarations relating to employment, consultancy, patents, products in development, or marketed products. The commercial affiliation, ANYMEDI, does not alter our adherence to PLOS ONE policies on sharing data and materials.

5. Response to reviewers

Reviewer #1-1: Abbreviations, such as VIBE, should be explained at the first time it appears in your article. You can put it in your title, but for both abstract and your manuscript, you should give the full text.

Answer: Thank you for your comment. We have given the abbreviation at the first appearance of the abstract and manuscript.

Reviewer #1-2. But Dixon and WE based fat suppression techniques are not based on the saturation of the fat. It is not proper to call them fat saturation techniques.

Answer: Thank you for your comment. We modified the term ‘fat saturation’ into ‘fat suppression’ throughout the manuscript.

Reviewer #1-3. Figures are quite low quality for the authors to see clearly. For example, the figure 2, the text in sub-figures can not be seen clearly at all.

Answer: The figure 2, and 3 are modified and reformatted. We have omitted some unnecessary sub-figures, which seems to be not informative. 

Review #1-4. The authors should give explanation for why fat suppression is necessary for the recon of the bone. Also the 3D osseous anatomy of the proximal femur extracted without fat suppression should be given as a baseline for comparison.

Answer: Thank you for your suggestion. We added the explanation for the reason why fat suppression was necessary in discussion section. “Suppression of the bone marrow was necessary to avoid chemical artifacts, which may be problematic especially in the femoral head area. The thickness of subchondral bone plate of femoral head measures below 0.9mm, which would be smaller than the size of a voxel.” This chemical artifact was noticed during the pilot study, and we concluded that fat suppression would be necessary when planning the study design. 

Review #1-5. Dixon based fat suppression are totally different from the WE in principle. So, it is quite interesting to figure out why they are equal in the discussion, from the medical physics view. It is expected that the authors would give more evidences, such as other reports.

Answer: Referring to the previous report of Kirchgesner et al that Dixon method was more effective in fat suppression and yielded higher signal to noise ratio compared to chemical shift-selective technique for 3D T1-weighted MR imaging, the authors chose WE method as a comparative arm for Dixon fat suppression method. There were no previous studies were found in the literature comparing these two fat suppression methods.

Review #1-6. References 14 and 15, seems some typos of the authors' name

Answer: Thank you for your comment. We have corrected the author’s name in the references.

---

## [Decision Letter · Decision Letter 1]

6 Apr 2021

Differences between 3D isovoxel fat suppression VIBE MRI and CT models of proximal femur osseous anatomy: A preliminary study for bone tumor resection planning

PONE-D-20-21489R1

Dear Dr. Chung,

We’re pleased to inform you that your manuscript has been judged scientifically suitable for publication and will be formally accepted for publication once it meets all outstanding technical requirements.

Kind regards,

JJ Cray Jr., Ph.D.

Academic Editor

PLOS ONE

Additional Editor Comments (optional):

Reviewers' comments:

Reviewer's Responses to Questions

**Comments to the Author**

1. If the authors have adequately addressed your comments raised in a previous round of review and you feel that this manuscript is now acceptable for publication, you may indicate that here to bypass the “Comments to the Author” section, enter your conflict of interest statement in the “Confidential to Editor” section, and submit your "Accept" recommendation.

Reviewer #1: All comments have been addressed

2. Is the manuscript technically sound, and do the data support the conclusions?

Reviewer #1: Yes

3. Has the statistical analysis been performed appropriately and rigorously? 

Reviewer #1: Yes

4. Have the authors made all data underlying the findings in their manuscript fully available?

Reviewer #1: No

5. Is the manuscript presented in an intelligible fashion and written in standard English?

Reviewer #1: Yes

6. Review Comments to the Author

Reviewer #1: Most of the issues were well addressed. However, there is one problem that the authors mentioned that the Dixon and WE are equivalent, but would be saying as no significant difference would be more accurate.

7. PLOS authors have the option to publish the peer review history of their article (what does this mean?). If published, this will include your full peer review and any attached files.

Reviewer #1: No

---

## [Editor Report · Acceptance letter]

21 Apr 2021

PONE-D-20-21489R1 

Differences between 3D isovoxel fat suppression VIBE MRI and CT models of proximal femur osseous anatomy: A preliminary study for bone tumor resection planning 

Dear Dr. Chung:

I'm pleased to inform you that your manuscript has been deemed suitable for publication in PLOS ONE. Congratulations! Your manuscript is now with our production department. 

Kind regards, 

on behalf of

Dr. JJ Cray Jr. 

Academic Editor

PLOS ONE